



# Vertical characterization of the dust fine and coarse particles during an intense Saharan dust outbreak over the Iberian Peninsula in springtime 2021

María-Ángeles López-Cayuela[1], Carmen Córdoba-Jabonero[1*], Diego Bermejo-Pantaleón[2,3], Michaël Sicard[4,5], Vanda Salgueiro[6,7], Francisco Molero[8], Clara Violeta Carvajal-Pérez[1], María José Granados-Muñoz[2,3], Adolfo Comerón[4], Flavio T. Couto[6,7], Rubén Barragán[8], María-Paz Zorzano[9], Juan Antonio Bravo-Aranda[2,3], Constantino Muñoz-Pocar[4], María João Costa[6,7], Begoña Artíñano[8], Alejandro Rodríguez-Gómez[4], Daniele Bortoli[6,7], Manuel Pujadas[8], Jesús Abril-Gago[2,3], Lucas Alados-Arboledas[2,3], Juan Luis Guerrero-Rascado[2,3]

[1]Instituto Nacional de Técnica Aeroespacial (INTA), Atmospheric Research and Instrumentation Branch, Torrejón de Ardoz, 28850-Madrid, Spain
[2]Andalusian Institute for Earth System Research (IISTA-CEAMA), 18006-Granada, Spain
[3]Dpt. Applied Physics, University of Granada (UGR), 18071-Granada, Spain
[4]CommSensLab, Dept. of Signal Theory and Communications, Universitat Politècnica de Catalunya (UPC), 08034-Barcelona, Spain
[5]Ciències i Tecnologies de l'Espai-Centre de Recerca de l'Aeronàutica i de l'Espai/Institut d'Estudis Espacials de Catalunya (CTE-CRAE/IEEC), Universitat Politècnica de Catalunya (UPC), 08034-Barcelona, Spain
[6]Institute of Earth Sciences (ICT) and Earth Remote Sensing Laboratory (EaRSLab), Évora, Portugal
[7]Department of Physics, University of Évora, 7000-671, Évora, Portugal
[8]Departamento de Medio Ambiente. Centro de Investigaciones Energéticas, Medioambientales y Tecnológicas (CIEMAT), Madrid, Spain
[9]Centro de Astrobiología (CSIC-INTA), Torrejón de Ardoz, 28850 Madrid, Spain

*Correspondence to*: Carmen Córdoba-Jabonero (cordobajc@inta.es)

**Abstract.** An intense and long-lasting Saharan dust outbreak over the Iberian Peninsula (IP) in springtime 2021 is studied in this work with the aim of investigating the optical and mass contribution of both the fine and coarse dust particles along their transport by crossing the IP from the south-west to the north-east. Five Iberian lidar stations (El Arenosillo/Huelva, Granada, Torrejón/Madrid and Barcelona in Spain and Évora in Portugal) were monitoring the transport and evolution of the Saharan dust particles across the Iberian Peninsula from 25 March until 7 April 2021. Particular meteorological conditions determined the aerosol scenario along the overall dust event, differing the first part of the event (25-31 March), with the strongest dust incidence on 29-31 March at the south and central stations, and 1 April at Barcelona, from the second one (1-7 April). Both the fine dust (Df) and coarse dust (Dc) components of the total particle backscatter coefficient (total dust, DD = Dc + Df) were separately derived. The dust plume was well-mixed and no significant differences were found in the vertical structure of both the Dc and Df particle backscatter coefficients. The vertical total dust mass concentration, and its Df/DD ratio, were estimated from the extinction profiles using the well-defined conversion factors for North African dust. From the beginning of the dust outbreak until 1 April, the vertical Df/DD mass ratio was nearly constant in time at each station and also almost constant in altitude with values ~ 10%. Regarding the columnar properties, the dust optical depth at 532 nm (DOD[532]), the relative mass loading and the centre-of-mass height were calculated, distinguishing the corresponding fine- and coarse-dust contributions. Firstly, the dust outbreak crossed the Iberian Peninsula from South-West to North-East and the mean DOD[532] was decreasing along that dust pathway, reporting values from SW to NE stations of 0.34 at El Arenosillo/Huelva, 0.28 at Granada, 0.20 at Évora, 0.28 at Torrejón/Madrid and 0.14 at Barcelona, although its Df/DD ratio kept almost constant (28-30%). On the other hand, a similar pattern was found for the total dust mass loading and its Df/DD ratio, i.e. mostly decreasing mean mass values were reported: 586 mg m[-2] (El Arenosillo/Huelva), 483 mg m[-2] (Granada), 332 mg m[-2] (Évora), 464 mg m[-2] (Torrejón/Madrid) and 248 mg m[-2] (Barcelona), being constant its Df/DD ratio (~ 10%) along the SW-NE dust pathway In addition, the episode-mean centre-of-mass height increased with latitude in overall, showing a high variability, i.e. being greater than 0.5 km at the southern sites (El Arenosillo/Huelva, Granada, Évora) and ~1.0 km at Torrejón/Madrid and Barcelona. However, despite the



relatively high intensity of the dust intrusion, the expected ageing of the dust particles was barely observed. This is on the basis that the Iberian Peninsula is relatively close to the Saharan dust sources and then, under certain dust transport conditions, any potential ageing processes in the dust particles remain unappreciated. It must be highlighted the different relative contribution of the fine dust particles to the total dust found for their optical properties (~ 30%), associated to the radiative effect of dust, with respect to the mass features (~ 10%), linked to air quality issues, along the overall dust event by crossing the IP. Finally, the methodology introduced, which is mainly based on the two-step POLIPHON algorithm, shows the relevance of the use of the polarized lidar measurements, mainly with elastic systems, for separating the aerosol properties for the dust fine and coarse particles as an added value to the simple one-step approach used for simple dust/non-dust separation.

## 1 Introduction

The impact of atmospheric aerosol particles on climate change is directly related to both their optical and microphysics properties as well as their spatiotemporal distribution (IPCC 2013). However, there are still large uncertainties associated with their aerosol direct and indirect radiative effects, mainly due to the change in the aerosol properties during the transport, the incomplete characterization of complex mixtures and the lack of information on cloud-aerosol interaction mechanisms (Stevens, 2015). The most abundant aerosol in the atmosphere is mineral dust, leading to worse air quality with harmful effects on human health (e.g., Diaz et al., 2017; Hashizume et al., 2020; Querol et al., 2019), specially under scenarios of extreme event (e.g., Euphrasie-Clotilde et al., 2021), with potential socio-economic impacts. Thus, the study of their variability on a global, regional and local scale is extremely useful to improve the understanding of atmospheric processes and model evaluation, and currently, the mineral dust is widely studied, becoming a remarkable scientific discipline (Ho et al., 2018; Middleton, 2017).

Global models estimated the total mineral dust emissions counting for 1000–3000 Tg yr$^{-1}$ (e.g., Zender et al., 2004) but more recent studies, based on a global dust model intercomparison, suggested that they may range from 500 to 4000 Tg yr$^{-1}$ (Huneeus et al., 2011). In particular, the Sahara desert is the main source of airborne mineral dust (Shao et al., 2011), representing half of the global mineral dust emissions, between 400 to 2200 Tg yr$^{-1}$ (Huneeus et al., 2011). Saharan dust can be transported to Europe under certain meteorological conditions (e.g., Salvador et al., 2014; Marinou et al., 2017; Russo et al., 2020; Couto et al., 2021), occasionally reaching rather high altitudes (up to 8 km height; e.g., Mona et al., 2006; Papayannis et al., 2008, Córdoba-Jabonero et al., 2021a; Sicard et al., 2022).

A great effort has been accomplished in the last decades in order to assess the mineral dust effect along the Mediterranean basin, focused on the dust vertical distribution by using ground-based micro-pulse lidars and advanced lidar systems belonging to EARLINET (European Aerosol Research Lidar NETwork; https://www.earlinet.org/; Pappalardo et al., 2014). Thus, several studies related to the study of dust optical and microphysical properties (e.g. Di Girolamo et al., 2012; Granados-Muñoz et al., 2016; Soupiona et al., 2019) and their impact on radiative forcing (e.g. Sicard et al., 2016; Valenzuela et al., 2017; Soupiona et al., 2020; Kokkalis et al., 2021) were performed. Recent studies show an increase in the frequency of Saharan outbreaks over Europe when compared with long-term records (Salvador et al., 2022; Sousa et al., 2019). Moreover, since extreme dust outbreaks are more and more frequently detected (Guerrero-Rascado et al., 2009; Mamouri et al., 2016; Cazorla et al., 2017; Solomos et al., 2017; Fernández et al., 2019) the WMO SDS-WAS (World Meteorological Organization Sand and Dust Storm Warning Advisory and Assessment System; https://public.wmo.int/en/our-mandate/focus-areas/environment/SDS) is devoted to research forecasting products from atmospheric dust models to contribute to risk reduction in many areas of societal benefit; and also, more recently, EARLINET has introduced an early warning system for atmospheric aerosol aviation hazards (Papagiannopoulos et al., 2020).

In particular in the Iberian Peninsula (IP), the arrival of Saharan dust intrusions is more frequently observed in springtime and summertime (e.g. Guerrero-Rascado et al., 2008; Córdoba-Jabonero et al., 2011; Obregón et al., 2015; Mandija et al., 2017;



Salvador et al., 2019; Córdoba-Jabonero et al., 2021a; López-Cayuela et al., 2021; Salgueiro et al., 2021; Abril-Gago et al., 2022), mainly at central and southern IP (Russo et al., 2020). An increase in the number of studies reporting severe and extreme events over the IP is also remarkable (Sánchez et al., 2007; Guerrero-Rascado et al., 2009; Preissler et al., 2011; Valenzuela et al., 2017), including the less and less extraordinary winter-time dust outbreaks (Cazorla et al., 2017; Córdoba-Jabonero et al., 2019; Fernández et al., 2019).

The aim of this work is to study an intense Saharan dust event occurring over the IP in springtime 2021. It was monitored by five Iberian lidar stations, which covered mostly the IP from SW to NE, leading to the analysis of the optical and microphysical properties of both the coarse and fine dust particles, once separated each of their contributions to the potential ageing processes along their transport by crossing the IP. The paper is structured as follows: the instrumentation used in each station and the methodology applied are described in **Sections 2** and **3,** respectively; the results and discussion are described in **Sections 4** and **5,** respectively; and, finally, the main conclusions are exposed in **Section 6**.

## 2 Monitoring stations and instrumentation

An intense Saharan dust outbreak was observed over the IP in spring 2021 from 25 March until 7 April, being monitored by five Iberian lidar stations. **Table 1** shows particular periods when the dust event was detected over each station (from SW to NE, by latitude): El Arenosillo/Huelva (ARN), Granada (GRA), Évora (EVO), Torrejón/Madrid (TRJ) and Barcelona (BCN). The relative position of those five stations and their corresponding geographical coordinates are shown in **Figure 1** and **Table 1**, respectively. Notice that they are located over a wide extension covering most of the IP. An overview of the lidar system deployed at each station is shown in **Table 2**.

On the one hand, polarized Micro-Pulse Lidars (P-MPL, v.4B, Droplet Measurement Technologies LLC, USA) are operating at El Arenosillo/Huelva (ARN), Torrejón/Madrid (TRJ) and Barcelona (BCN). The P-MPL system is a one-wavelength elastic lidar, with a relatively high pulse repetition frequency (2500 Hz) using low-energy (~7μJ) Nd:YVO$_4$ laser at 532 nm, including polarization capabilities. It operates in an automatic and full-time continuous mode (24/7). A dead-time correction was applied following the manufacturer's instructions and laboratory calibrations of the detector (Campbell et al., 2002). Dark-count and after-pulse correction measurements are performed on a monthly basis (Campbell et al., 2002; Welton and Campbell, 2002). The MPL total range-corrected signal (RCS) and volume linear depolarization ratio (VLDR) were determined according to Flynn et al. (2007) by using in-house, well-validated data processing (Córdoba-Jabonero et al., 2018; Sicard et al., 2020; Córdoba-Jabonero et al., 2021b). Both the RCS and the VLDR ($\delta^v$) were obtained as hourly averaged profiles to increase the signal-to-noise ratio. Additionally, vertically-resolved particle backscatter coefficients ($\beta_p$) were retrieved by using the Klett-Fernald algorithm (Fernald, 1984; Klett, 1985) by assuming a fixed lidar ratio of 50 sr; the particle linear depolarization ratio ($\delta_p$) profiles were calculated from both the $\beta_p$ and $\delta^v$ ones.

On the other hand, multi-wavelength Raman lidar (RL) systems are deployed at EVO and GRA stations, forming part of ACTRIS/EARLINET network. Those RL operate at three elastic wavelengths (355, 532 and 1064 nm), two Raman channels (355 and 532 nm) and one polarization-sensitive channel (532 nm), using high energy (~ 65-400 mJ) Nd:YAG laser and relatively low pulse repetition frequency (10-20 Hz). Specifically, the RL deployed at GRA station is a LR331D400 system (Raymetrics S.A., Greece) described in detail in Guerrero-Rascado et al. (2008, 2009), and a Polly$^{TX}$ system (Baars et al., 2016; Engelmann et al., 2016) is operative at EVO station (Preissler et al., 2011). The Single Calculus Chain (SCC; D'Amico et al., 2016; Mattis et al., 2016), which is the EARLINET standardised tool for lidar data processing, is used to retrieve the vertical $\beta_p$ and $\delta_p$ profiles from the elastic RL measurements. Those are retrieved at 532 nm only for comparison with the same P-MPL-derived profiles, fixing a lidar ratio of 50 sr in the SCC algorithm. A complete review of the lidar techniques using elastic, Raman and polarization-sensitive channels can be found in Comerón et al. (2017).





### 3 Methodology

#### 3.1 Transport pathway and synoptic situation

The origin and pathway of the dust events affecting the IP were analysed by using several models to accurately determine the
spatial and temporal coverage of the dusty event over the stations (see **Table 1**). On one hand, a first and general insight of the
geographical allocation of the aerosol plumes was obtained by means of the added-value aerosol optical depth from Terra/Aqua
MODIS satellite (https://modis.gsfc.nasa.gov/about/; images not shown) and also from Meteosat SEVIRI image RGB
composites tailored to monitor the evolution of dust storms (https://view.eumetsat.int/; images not shown). On the other hand,
an overview of the synoptic situation favouring the arrival and spreading of these dust particles was performed, by using the
NCEP/NCAR (National Centers for Environmental Prediction and National Center for Atmospheric Research) reanalysis.
Specifically, images of the mean geopotential height (m) are shown. Those images are provided by the NOAA Physical
Sciences Laboratory (http://psl.noaa.gov/data/composites/day/; last access: 11 April 2022).

Additionally, the PySPLIT HYSPLIT Toolbox (Warner, 2018) has been used to identify the source area of the dust particles
as observed over the stations. This is a package that contains functions and classes to automatically generate trajectories from
the NOAA Air Research Laboratory's HYSPLIT model (Hybrid Single-Particle Lagrangian Integrated Trajectory model
version 4; Rolph et al., 2017; https://ready.arl.noaa.gov/HYSPLIT.php). In particular, a total of 456 single five-day back-
trajectories of air masses have been computed for each station and day of the dust event, including their vertical extent from
500 to 9500 m height (500-m step). That trajectory assembly is analysed in terms of the percentage of back-trajectory points
overpassing the Saharan area. It is assumed that dust particles are potentially transported by those air masses if at least one
point of their pathway is crossing the Sahara; otherwise, the trajectory is discarded. The Global Data Assimilation System
database (GDAS, ftp://www.arl.noaa.gov/puv/archives/gdas1; last access: 12 November 2021; spatial resolution of 1º x 1º
every 3h) was used to feed the HYSPLIT model.

#### 3.2 Separation of the optical and mass properties

In this work, the Polarisation LIdar PHOtometer Networking (POLIPHON) approach (Mamouri and Ansmann, 2014, 2017) is
used for separating the optical properties (backscatter and extinction) of aerosol mixtures. This method, which is applied to
elastic polarized lidar measurements, separates particle components with clearly different depolarization ratios. Specifically,
the two-step POLIPHON approach was applied, using the total $\beta_P$ and $\delta_P$ profiles at 532 nm to discriminate the three
components of a dusty mixture, namely the dust fine (Df) and coarse (Dc), and non-dust (ND) components. The latter is
assumed to be representative of background fine aerosols with low depolarizing particles ($\delta_P \sim 0.05$). Note that $\beta_P$ is an
extensive parameter and, therefore, $\beta_P = \Sigma_i \beta_i(z)$, where $\beta_i$ (i = Df, Dc, ND) are the corresponding backscatter coefficients for
Df, Dc, and ND components, respectively, and $z$ denotes the height dependence. Hence, once the single backscatter coefficient
($\beta_{i,}$) is separated, the profiles of dust extinction coefficient for each component ($\alpha_i$) can be obtained by considering the specific
particle lidar ratios at 532 nm for each component i (**Table 3**). Therefore, the total particle extinction coefficient $\alpha_P$ is calculated
as $\alpha_P = \Sigma_i \alpha_i = \Sigma_i LR_i \beta_i(z)$, where i = Df, Dc, ND. More details on the extinction retrieval by components can be found in
Córdoba-Jabonero et al. (2018, 2021a). The total dust (DD) extinction coefficient can be obtained as the sum of the extinction
coefficients for the Df and Dc components. Thus, the dust optical depth at 532 nm (DOD[532]) for each dust component DOD$_i$[532],
i = Df, Dc is obtained by height-integration of each $\alpha_i(z)$. The assumed specific values of $\delta_P$ and lidar ratio used in the
POLIPHON retrieval and the uncertainties estimated for each variable can be found in **Table 3** and **Table 4** (Mamouri and
Ansmann (2017); Ansmann et al. (2019)), respectively.
Additionally, the dust mass features along the dust event are derived from the extinction profiles for the Dc and Df components,
as described in Córdoba-Jabonero et al. (2019). Indeed, the vertical profile of the total dust mass concentration (m$_{DD}$) can be
calculated as follows:



$$m_{DD}(z) = m_{Df}(z) + m_{Dc}(z) = d_p (c_{vDf} \alpha_{Df}(z) + c_{vDc} \alpha_{Dc}(z)), \tag{1}$$

where $d_p$ is the dust particle density (g cm$^{-3}$) and $c_v$ is the volume-to-extinction conversion factor (10$^{-12}$ Mm) for each dust component (Dc, Df). Their assumed values (and references) are shown in **Table 3**, and their uncertainties can be found in **Table 4**. In addition, the total dust mass loading (M$_{DD}$, in g m$^{-2}$) can be obtained by the height-integration of the mass concentration profiles as follows:

$$M_{DD} = M_{Df} + M_{Dc} = \sum_z m_{Df}(z) \Delta z + \sum_z m_{Dc}(z) \Delta z, \tag{2}$$

where $\Delta z$ is the vertical resolution of the profiles for each lidar system (see **Table 2**). The relative mass contribution of each component to the total dust mass loading ($M^i$, in %, with i = Dc, Df) is also calculated, and it is expressed as:

$$M^i = 100 \frac{M_i}{M_{DD}}. \tag{3}$$

Finally, the centre-of-mass (CoM) height ($Z_{CoM}$) is also estimated in order to provide a measure of the vertical mass impact of each component (Córdoba-Jabonero et al., 2019), which is expressed as:

$$Z_{CoM}^i = \frac{\sum_k z_k m_i(z_k) \Delta z}{\sum_k m_i(z_k) \Delta z}, \tag{4}$$

where $m_i$ are the mass concentration profiles for each i component (see **Eq. 1**, i = Df, Dc) and $z_k$ is the height at the k-step as defined by the vertical resolution of each lidar system (**Table 2**).

## 4 Results

### 4.1 Synoptic scenario

The meteorological situation is described by using the NCEP/NCAR Reanalysis data in order to check the conditions favouring the advection injection of Saharan dust over the IP (Salvador et al., 2014; Diaz et al., 2017; Couto et al., 2021). On the one hand, the meteorological scenarios are described by using daily pictures of Geopotential Height Composite Mean at 700 hPa (around 3000 m a.s.l.) on particular days (**Fig. 2**). On the other hand, the geopotential at 500 hPa (around 5500 m a.s.l.) was selected on two critical days, namely 29 March (**Fig. 3a**) and 1 April (**Fig. 3b**; vertical temperature gradient), when dust layers were observed on relatively high altitudes at central and north-eastern stations.

Before the dust intrusion arrival at the IP, the meteorological situation in the region was governed by the North Atlantic Anticyclone (**Fig. 2a**). From 25 March (**Figs. 2b-2c**) the Geopotential Height map at 700 hPa shows a low-pressure system (L1) centred over the Canary Islands and leading to a southern/southeastern (S/SE) Circulation Weather Type (CWT). The highest probability of dust occurrence during the meteorological spring season (March-April-May) was 60-80%, in particular, under the S/SE CWT (Russo et al. 2020). Under this synoptical situation, a southern flow was created, favouring the arrival of air masses from Northern Africa, making the dust advection feasible and carrying dust loads through the IP.

From 30 March (**Figs. 2d-2e**), another low-pressure system (L2) moved down from higher latitudes, and the gradient between the high-pressure and the low-pressure systems intensified (**Fig. 2e**), favouring also the increase in the wind magnitude. The low-pressure system (L2) moved toward the west IP from 31 March to 1 April (**Figs. 2e-2f**). This low pressure was accompanied by a closed upper-level low-pressure system (500 hPa, **Fig. 3a**) isolated from the general atmospheric circulation, which led to convection and instability. The convection led to a rise in the air parcel coming from North Africa transporting dust particles. Moreover, this atmospheric instability induced vertical movements of the air parcel, rising dust to high heights (see **Sect. 3.3**), favouring the formation of high-level thick ice clouds that gradually affected most of the IP from SW to NE, between the afternoon of 31 March and the night of 1 April. These vertical motions can also be related to the presence of the diffluent flow at 250 hPa and its relationship with the low-pressure system whose distribution extended up to this level and from 700 hPa (**Fig. 3**; Geopotential Field at 250 hPa not shown). Particularly, Yang et al. (2022) found that the presence of dust in the upper troposphere (above 6 km m.s.l.) is most prominent in spring in the northern hemisphere, due to the unique


combination of the different annual cycles of the westerly jet and the presence of dust at 4-6 km m.s.l. For the African source, trough lifting is the leading uplift mechanism associated with a large temperature gradient in the mid-latitude and frequent synoptic cyclones.

The second low-pressure system (on the surface and at elevated levels) crossed the IP from 1 April to 3 April (**Figs. 2f-2g**), favouring precipitation that advanced from the west towards the centre of the IP. Meteosat satellite images confirm the presence of low-level clouds (https://view.eumetsat.int/; images not shown here) and NCEP Climate Forecast System Reanalysis displayed precipitation (data not shown). This precipitation could have produced wet scavenging of dust aerosols and, consequently, the removal of these particles from the atmosphere at ARN, GRA, EVO and TRJ stations (1-3 April).

Nevertheless, the cyclonic wind system (L2) could have re-circulated dust particles again to the south IP, since the airflow came from NE IP, which was under dusty conditions. Finally, on 5 April, L2 had dissipated, and a northerly flow was verified over the northeast of the Peninsula at higher levels.

To sum up, the event mobilised a high amount of dust from 25 March to 7 April, and even with the dissipation of the low pressure (L2, 5 April), some dust remained suspended in the atmosphere. Thus, dust was still observed on 5 April at BCN,

which was transported by the predominant atmospheric circulation on that day, namely northerly winds above 500 hPa (see **Sect. 4.2**).

Furthermore, the percentage of HYSPLIT back-trajectories that comes from the Sahara desert area per day and lidar station, thus potentially carrying dust particles, are shown in **Fig. 4**. From the beginning of the episode, the percentage of Saharan back-trajectories was gradually increasing, finding maximum values (75-100%) between 28 and 31 March. It should be noted

that the maximum incidences shifted over time with increasing latitude of the lidar station, revealing the SW-NE crossing of the dust plumes over the IP. From 1 to 3 April, the percentage of back-trajectories decreased significantly, down to values of 10%, at ARN (**Fig. 4e**), GRA (**Fig. 4d**), EVO (**Fig. 4c**) and TRJ (**Fig. 4b**), coinciding with the low-pressure system conditions explained above. From 3 to 4 April, the incidence slightly increased again (**Figs. 4b-4e**), reaching percentages of 25-35%, still leading to dust particle advection. Finally, the number of trajectories from the Saharan source was almost zero from 5 April

on.

Therefore, regarding an overview of the dust plume behaviour by crossing through the IP, the desert dust plume entered the IP from the SW on 25 March (ARN, EVO), one day later reached the SE (GRA) and centre (TRJ) and, finally, the eastern part of the IP (BCN) on 28 March (**Fig. 1, Table 1**). Finally, it should be noted that the dust event was profusely cloudy over the five Iberian lidar stations, preventing some lidar retrievals. Hence, several gaps can be found in the inversion dataset, together with

the periods with no lidar measurements (see **Fig. 5**).

## 4.2 Optical properties: dust extinction and depolarization profiles

Potential changes in the optical properties of dust particles during the dust outbreak over the IP are analysed, especially regarding the dust extinction and depolarization profiles.

**Figure 5** shows an overview of the temporal evolution of the dust intrusion crossing the IP in terms of the vertical dust $\beta_{DD}$

profiles over the five lidar stations. At the southern stations, the dust plume was detected at low altitudes (top of the dust plume below 3 km a.s.l.; ARN and EVO on 25 March, GRA on 26 March; henceforth the term 'a.s.l.' is omitted for simplicity, considering all heights above sea level). On 27 March, due to atmospheric instability (see **Sect. 4.1**), the dust intrusion ascended, maintaining its top relatively constant below 6 km height as time progressed (27-31 March). From 1 April, the dust plume progressively descended, staying below 3 km until the end of the dust episode. At TRJ, the dust plume was detected

below 4 km on 26 March. However, it ascended at the end of the day, reaching a maximum altitude of 8 km. On 28 March, the top of the dust layer reached 10 km height. On the successive days, its maximum varied between 6-8 km, reaching again 10 km height on 31 March, and then progressively descending until 4 km height at the end of the episode. At BCN station, the vertical dust structure was substantially stratified during the overall dust occurrence. On 28 March, the dust plume was below



4 km height, but one day later, it ascended to altitudes of 8 km until the end of the dust episode and even reached a maximum
altitude of 10 km height several times along its pathway.

It should be noted the atypical, very high altitudes that the top of the dust plume reached over the centre and north-eastern stations. However, those top heights have been found for dust at mid-latitudes by other authors with ground-based lidar instrumentation. For instance, a 3-year statistic of Saharan dust intrusions over the central Mediterranean showed the detection of dust particles between 1.8 and 9 km height, and the top of the dust layer ranged between 3.1 to 8.9 km, with an average
value of $5.9 \pm 1.2$ km (Mona et al., 2006). Moreover, Papayannis et al., (2008) studied 2 years of Saharan dust intrusions over the south and south-east Europe, finding the top of the dust plume at maximum values of 9-10 km height. Particularly in BCN, averaged top dust layer values of $3.6 \pm 1.6$ km were found, although the top height ranged between 1.2 and 9.0 km.

**Figure 6** shows the height-resolved $\beta_p$ and both the $\beta_{Df}$ and $\beta_{Dc}$ components, together with the $\delta_p$ profiles, for representative dates, including the beginning of the dust event (**Fig. 6a**), the day of the maximum incidence (**Fig. 6b**) and the end of the event
(**Fig. 6c**) for each lidar station. In general, the height-resolved $\beta_{Df}$ and $\beta_{Dc}$ components presented the same structure in height, indicating that they were well mixed in the dust layer (**Fig. 6**). At the southern stations (i.e. ARN, GRA, EVO), the dust plume followed the same pattern. At the beginning of the event, mineral dust was detected at low altitudes (below 3.0 km). On 27 March, the dust intrusion ascended due to the atmospheric instability, reaching up to 6.0 km. Maximum $\beta_p$ values of around 5, 3 and 5 $Mm^{-1} \cdot sr^{-1}$ for ARN, GRA and EVO, respectively, were observed. The greatest incidence of the dust intrusion was
observed on 29-31 March, showing a pronounced dusty structure with a predominance of Dc particles and extending from the surface up to 8.0 km height. In particular, at ARN (GRA) station, a well-defined dust layer was found at 2.0-5.0 (3.0-5.0) km, with the maximum peaks of $\beta_p$, $\beta_{Dc}$ and $\beta_{Df}$ ranging between 3.0-8.0 (2.0-6.0), 2.0-6.0 (1.0-4.0) and 0.5-3.0 (1.0-1.5) $Mm^{-1} \cdot sr^{-1}$, respectively. Specifically, the dust layer for the dust cases shown in **Figure 6b** presented a maximum peak at 3.0-4.5 (4.0-4.5) km, with averaged $\beta_p$, $\beta_{Dc}$ and $\beta_{Df}$ values of $6.2 \pm 0.2$ ($5.0 \pm 0.5$), $4.0 \pm 0.1$ ($3.4 \pm 0.6$) and $1.8 \pm 0.1$ ($1.2 \pm 0.5$) $Mm^{-1} \cdot sr^{-1}$
at 3.0-4.5 (4.0-4.5) km height. In the case of EVO station, the pronounced dust layer was found at 2.0-3.0 km with the maximum peaks $\beta_p$, $\beta_{Dc}$ and $\beta_{Df}$ ranging between 2.0-7.0, 1.0-3.0 and 0.5-1.0 $Mm^{-1} \cdot sr^{-1}$, respectively. In the dust cases, as shown for EVO in **Figure 6b,** the maximum peak was found at 1.0-1.5 km, showing mean $\beta_p$, $\beta_{Dc}$ and $\beta_{Df}$ values in that dust layer of $5.3 \pm 0.2$, $1.7 \pm 0.1$ and $0.7 \pm 0.1$ $Mm^{-1} \cdot sr^{-1}$. Regarding the days of the maximum incidence (29-31 March), the peak of the $\beta_p$ values ranged between similar values at the three southern stations. However, the maximum peak of $\beta_{Dc}$ was 1.5 and 2 times greater
at ARN with respect to GRA and EVO stations, respectively, and the maximum $\beta_{Df}$ values were 2 times greater at ARN station, with respect to the other two southern stations. From 1 April on, the dust plume progressively descended down to 3.0 km height until the end of the dust episode for those southern stations. In addition, the dust incidence was weaker, with maximal $\beta_p$ values of ~2.5, 1.0 and 2.5 $Mm^{-1} \cdot sr^{-1}$ for ARN, GRA and EVO, respectively.

Overall, at the beginning of the episodes, $\delta_p$ showed values of around 0.30, 0.25-0.30 and 0.20-0.30 at ARN, GRA and EVO
stations, respectively. During the greatest incidence (29-31 March), $\delta_p$ showed values around 0.30 at those three stations, indicating a clear predominance of Dc particles; then $\delta_p$ decreased until values of 0.15-0.25 at the end of the episode. These values agree with those found for pure dust particles over Morocco and Cape Verde during the SAMUM and SAMUM-2 (Saharan Mineral Dust Experiment) campaigns, ranging from 0.23 to 0.31 (Freudenthaler et al., 2009; Gross et al., 2011). Other studies reported similar values at the stations considered in this work. In particular, $\delta_p$ values of 0.26-0.31 and 0.15-0.19
were found at GRA and EVO, respectively, during an extreme dust event over the IP in wintertime (Fernández et al., 2019), meanwhile $\delta_p$ of 0.26 for pure dust and 0.28 for polluted dust were obtained for a dust outbreak occurred in the summertime at GRA (Soupiona et al., 2019). Finally, a study of a mixed-event with smoke and dust showed values ranging from 0.24 to 0.28 for pure dust at EVO (Salgueiro et al., 2021).

By examining the dust intrusion at the central IP, the dust plume as observed at TRJ was firstly detected below 4.0 km on 26
March. However, it ascended up to a maximum altitude of 10.0 km at the end of the day. On successive days, the top of the dust plume varied between 6.0-8.0 km height, reaching 10.0 km several times. Particularly, the greatest incidence was found



on 29-31 March, with $\delta_p$ values of 0.25 on average, and maximum $\beta_p$, $\beta_{Dc}$ and $\beta_{Df}$ values of 2.0-18.0, 1.0-9.0 and 0.5-4.0 Mm$^{-1}\cdot$sr$^{-1}$, respectively. In the dust cases shown in **Figure 6b**, the maximum peak was found at 4.0-4.5 km, showing $\beta_p$, $\beta_{Dc}$ and $\beta_{Df}$ values, on average, of 5.2 ± 0.1, 2.7 ± 0.1 and 1.2 ± 0.1 Mm$^{-1}\cdot$sr$^{-1}$, together with an averaged $\delta_p$ of 0.21 ± 0.01. It should be

noted the highest $\beta_p$ peaks are found in this station on 31 March (around 09-13 UTC), with hourly values ranging 16-18 Mm$^{-1}\cdot$sr$^{-1}$ at 4.0-5.0 km height. These values are 3 times higher than the maximum values found at the southern stations. Moreover, the dust backscatter component ($\beta_{DD}$) ranged between 10-12 Mm$^{-1}\cdot$sr$^{-1}$ with DOD$^{532}$ ~1.0. Similar results to those reported in Córdoba-Jabonero et al., (2019) are found for $\beta_{DD}$ with peaks around 10-15 Mm$^{-1}\cdot$sr$^{-1}$ over 3 km height (and finding a more pronounced dust layer between 2.0-5.0 km), and DOD$^{532}$ values around 1.6. In this study, although the peak $\beta_{DD}$ is 20% lower

with respect to the previous one, the DOD$^{532}$ is around 40% lower, the thickness is smaller. Finally, the dust plume decayed from 8.0 to 4.0 km height from 1 April until the end of the dust episode, with lower $\beta_p$ and $\delta_p$ values ranging between 1-3 Mm$^{-1}\cdot$sr$^{-1}$ and 0.15-0.25, respectively. Similarly, $\delta_p$ ~ 0.25 were reported at Madrid (20 km far from TRJ) (Fernández et al., 2019) and around 0.30 at TRJ (Córdoba-Jabonero et al., 2019).

Finally, the dust situation at the NE of the IP, monitored at BCN, was less intense with respect to that observed at the other

lidar stations. On 28 March, the dust plume was found between 2.0-3.0 km height, showing $\beta_p$ values ranging between 1.0-1.5 Mm$^{-1}\cdot$sr$^{-1}$, which are lower than those found at the other stations. However, $\delta_p$ values also indicated a relatively high predominance of Dc particles ($\delta_p$ around 0.30-0.35). One day later, the dust conditions at BCN were more complex. At the beginning of the day, two layers were found at 2.0-3.0 and 9.0-10.0 km, with peak $\beta_p$ values around 0.3 and 0.2 Mm$^{-1}\cdot$sr$^{-1}$, and $\delta_p$ values between0.31-0.35 and ~0.25, respectively. At the end of the day, three layers were found at 1.0-2.0, 4.0-7.0 and 8.0-

10.0 km, with $\beta_p$ peak values around 0.9, 0.5 and 0.1 Mm$^{-1}\cdot$sr$^{-1}$, and $\delta_p$ values around 0.33, 0.28 and 0.25, respectively. During the successive days, the dust plume followed the same behaviour, reaching its maximum incidence on 1 April. Looking at the dust cases as shown in **Figure 6b**, two different dust layers were found, ranging from 2.0-6.0 and 6.0-8.0 km height, with averaged $\beta_p$ values of 1.8 ± 0.4 and 0.6 ± 0.2 Mm$^{-1}\cdot$sr$^{-1}$, and $\delta_p$ values of ~ 0.30 and ~ 0.35, respectively. Moreover, $\beta_{Dc}$ ($\beta_{Df}$) showed values of 1.4 ± 0.3 (0.5 ± 0.1) Mm$^{-1}\cdot$sr$^{-1}$ and 0.5 ± 0.1 (< 0.1) Mm$^{-1}\cdot$sr$^{-1}$, for both dust layers, respectively. Until the

end of the dust episode, the incidence was decreasing, with $\beta_p$ and $\delta_p$ values of between 0.5-2.0 and 0.20-0.30, respectively. Similar $\delta_p$ values were found at the BCN station ranging from 0.23 to 0.31 (Córdoba-Jabonero et al., 2019, 2021a; Fernández et al., 2019).

Regarding the dust impact on the overall atmosphere, columnar extinction values (i.e., the dust optical depth at 532 nm, DOD$^{532}$) were also estimated. **Figure 7** shows the DOD$^{532}$ evolution in terms of the daily-averaged DOD$^{532}$ ($\overline{DOD^{532}}$) and its

fine-to-total ratio (Df/DD DOD$^{532}$, in %; it will be denoted as ftr_DOD afterward) along the dust episode for the five lidar stations ($\overline{DOD^{532}}$ values are shown in **Table S1**). At the southern stations, $\overline{DOD^{532}}$ progressively increased reaching values of 0.77 ± 0.13, 0.15 ± 0.03 and 0.33 ± 0.05, respectively, at ARN, GRA and EVO on 27 March (note that SW ARN and EVO stations present higher DODs) (**Figs. 7c-7e**; **Table S1**). On 28 March, $\overline{DOD^{532}}$ decreased overall, showing values of 0.17 ± 0.03, 0.12 ± 0.03 and 0.15 ± 0.06, respectively (note that this decrease was less severe at GRA as well). However, due to the

enhancement of the gradient between the high- and low-pressure systems that occurred on that day (see **Sect. 3.1**), an intensification of the dust outbreak over the PI was observed lately on 29-31 March, being more pronounced over ARN and GRA stations. Specifically, $\overline{DOD^{532}}$ (ftr_DOD) reached rather high (low) values: 1.02 ± 0.26 (25.5%) at ARN and 0.59 ± 0.04 (28.3%) at GRA. Moreover, the hourly DOD$^{532}$ values of 1.30 and 0.65 peaked at ARN and GRA, respectively. The anomalies (i.e., the difference between the mean of the episode and the daily mean) of the ftr_DOD are -3.7 (ARN) and -5.8 (GRA) on

29 March, highlighting the dominance of coarse dust particles during the highest dust incidence, as expected. In the case of EVO station, the intensification of the dust outbreak is, on average, less pronounced, reaching maximum $\overline{DOD^{532}}$ (ftr_DOD) values of 0.36 ± 0.11 (30.6%) on 30 March, and an hourly DOD$^{532}$ peak of 0.60. On 1 April, the weather conditions avoided the dust observations over the southern stations, and apparently, wet deposition occurred. Later on (2-7 April), the dust incidence was significantly weaker, finding maximal $\overline{DOD^{532}}$ (ftr_DOD) values of 0.15 ± 0.04 (33.3%), 0.06 ± 0.01 (33.3%)



and 0.14 ± 0.08 (28.6%) at ARN, GRA and EVO stations, respectively. $\overline{DOD^{532}}$ was 7, 5 and 2 times lower than the maximal dust incidence for the first part of the dust event (29-31 March). For the overall event, averaged $\overline{DOD^{532}}$ (ftr_DOD) was 0.34 ± 0.35 (29.4%), 0.28 ± 0.22 (28.6%) and 0.20 ± 0.11 (30.0%) at ARN, GRA and EVO, respectively.

Unfortunately, the meteorological conditions at the central IP prevented lidar measurements at TRJ on 27 March and 2 April. The dust incidence shows the same behaviour as in the southern stations; the $\overline{DOD^{532}}$ (ftr_DOD) progressively increased (**Fig. 7b, Table S1**), reaching high values between 29 and 31 March, with maximal values of 0.6 ± 0.03 (28.6%) on 29 March. By comparing those values with the highest daily DOD$^{532}$ as found at the southern stations (i.e., on 29 March at ARN and GRA), $\overline{DOD^{532}}$ was 1.5 times lower than that observed at the ARN station, and no significant differences with respect to GRA station, and neither in the ftr$_{DOD}$ for both stations. Notice that the hourly DOD$^{532}$ peaked on 31 March (DOD$^{532}$ = 0.88) at TRJ. For the entire dust episode, the averaged $\overline{DOD^{532}}$ (ftr_DOD) was 0.28 ± 0.22 (28.5%).

Finally, the dust situation at the NE of the IP, as observed at the BCN station (**Fig. 7a, Table S1**), show that the greatest incidence was observed on 1 April, when $\overline{DOD^{532}}$ reached 0.27 ± 0.20, being the ftr_DOD 25.9%. Taking as reference the highest incidence observed in the southern stations (on 29 March), $\overline{DOD^{532}}$ was almost 4 and 2 times lower than those values found at ARN and GRA, respectively. The averaged $\overline{DOD^{532}}$ (ftr_DOD) for the overall dust event was 0.14 ± 0.08 (28.6%). In general, along the overall dust event, $\overline{DOD^{532}}$ decreased as latitude increases, with mean values from 0.34 ± 0.35 as found at ARN to 0.14 ± 0.08 at BCN station, indicating the decay of the dust incidence by overcrossing through the IP from SW (e.g. at ARN) to NE (BCN). However, the ftr_DOD did not show significant differences (around 29%), showing a uniformly well-mixed state of the Dc and Df particles along their pathway over the IP.

### 4.3 Mass-related properties: mass concentration and centre-of-mass height

In order to assess the dust amount injected into the atmosphere that crossed the IP, the dust mass concentrations were derived at the five lidar stations (see **Sect. 3.2**). The evolution of the vertical extent of the Dc and Df components is analysed in terms of their relative mass contribution to the vertical total dust mass concentration.

**Figure 8** shows the mass concentration profiles ($m_{DD}$, in µg m$^{-3}$) and their fine-to-total (Df/DD) ratio (ftr_$m_{DD}$, %) for particular dusty cases, which correspond to those as shown in **Figure 6**. They illustrate the beginning of the dust event (**Fig. 8a**), the day of the maximum incidence (**Fig. 8b**) and the final of the event (**Fig. 8c**) for the five Iberian lidar stations. It is worth highlighting that, from the beginning of the event to the maximum incidence days, the ftr_$m_{DD}$ was nearly similar at all the stations, being almost constant in altitude, with values ~ 10%, except for BCN (see **Fig. 8b**) that showed lower values (~ 5%) at higher altitudes (6.0-8.0 km). At the end of the dust outbreak, the ftr_$m_{DD}$ followed the same tendency, except at TRJ and BCN where it increased (~15%) for the dust layers below 4.0 km height.

As it would be expected, the altitudes where the maximal $m_{DD}$ values were found are coincident with those where the maximums of $\beta_p$ were observed (**see Sect. 4.2**). At the beginning of the dust intrusion (25-27 March), the dust signature peaked with $m_{DD}$ values ranging 50-500, 80-260 and 150-500 µg m$^{-3}$ at ARN, GRA and EVO stations, respectively. On 28 March, the $m_{DD}$ decreased down to 90-150, 70, and 80-200 µg m$^{-3}$, respectively. For the period of the greatest incidence (29-31 March), maximum peaks of 200-750, 120-500 and 50-500 µg m$^{-3}$ were found at ARN, GRA and EVO. Finally, $m_{DD}$ ranged in 50-300 µg m$^{-3}$, 50-60 µg m$^{-3}$ and 30-300 µg m$^{-3}$ for the successive days over those stations. In short, at the beginning of the episode, the maximums of $m_{DD}$ were 50% smaller values at GRA with respect to those at ARN and EVO. However, GRA and EVO showed similar $m_{DD}$ peaks during the greatest incidence period, being around 40% lower than those found at ARN station. For the second part of the episode, ARN and EVO showed similar m$_{DD}$ values, whereas they were 5 times lower at GRA. In the case of TRJ, $m_{DD}$ peak values ranged in 30-350 µg m$^{-3}$, progressively increasing up to 1000 µg m$^{-3}$ on 31 March (9-13 UTC) and decreasing down to 15-260 µg m$^{-3}$ from 1 April to the end of the dust outbreak (5 April). This highest value (1000 µg m$^{-3}$) is 25% greater than those found at the southern ARN station (750 µg m$^{-3}$). By comparing with other results as reported at



TRJ (Córdoba-Jabonero et al., 2019; see **Sect. 4.2**), similar values are found, ranging from 900 to 1000 µg m$^{-3}$ and with similar β$_{DD}$ values. Finally, at the BCN station, the $m_{DD}$ peaked with around 40-130 µg m$^{-3}$ on 28-31 March, increasing up to 40-400 µg m$^{-3}$ on 1 April and decreasing down to 35-300 µg m$^{-3}$ from 2 to 5 April. The maximum m$_{DD}$ was 45%, 20%, 20% and 60% smaller with respect to those values found for ARN, GRA, EVO and TRJ. In order to assess the incidence of this dust outbreak,

these results can be compared with those derived for dust particles observed in West Europe. Binietoglou et al. (2015) performed an analysis of dust concentration profiles by using systematic observations of dust events at ten ACTRIS/EARLINET stations. The peak of the mean $m_{DD}$ ranged from 30 to 80 µg m$^{-3}$, and, thus, the maximum $m_{DD}$ values for the strongest dust incidence period as observed at the IP stations were 9 (ARN), 6 (GRA, EVO), 13 (TRJ) and 5 (BCN) times higher than the averaged values shown in Binietoglou et al. (2015).

Moreover, the dusty situation over the IP was also analysed examining the overall atmospheric dust impact in terms of the total dust mass loading ($M_{DD}$, in g m$^{-2}$; see **Eq. 3**) for the Dc and Df particles and their centre-of-mass height ($Z_{CoM}$, in km; see **Eq. 4**), as a measure of the height where the dust incidence is enhanced. **Figure 9** shows the evolution of the hourly M$_{DD}$ throughout the dusty episode together with their daily-averaged values ($\overline{M_{DD}}$), in addition to the hourly and daily dust fine-to-total mass loading ratio (ftr_M$_{DD}$, in %).

As expected, at the southern stations (ARN, GRA, EVO), $\overline{M_{DD}}$ (daily ftr_M$_{DD}$) (**Figs. 9c-9e, Table S2**) progressively increased (decreased) until reaching maximum (minimum) values on 29-30 March of 1809 ± 396 mg m$^{-2}$ (8.5%), 1018 ± 96 mg m$^{-2}$ (9.8%) and 605 ± 183 mg m$^{-2}$ (10.6%), respectively), and decreased later on. For the second part of the dust event, when the dust incidence was getting weaker, maximal $\overline{M_{DD}}$ (daily ftr_M$_{DD}$) values of 256 ± 63 mg m$^{-2}$ (10.9%), 82 ± 2 mg m$^{-2}$ (11.0%) and 232 ± 136 mg m$^{-2}$ (10.8%) were found at ARN, GRA and EVO, respectively. These $\overline{M_{DD}}$ values were, respectively, 7, 12

and 3 times lower with respect to those found for the first part of the event. Regarding the TRJ station, $\overline{M_{DD}}$ (daily ftr_M$_{DD}$) progressively also increased (**Fig. 9b, Table S2**), being the maximum values as observed on 29 March of 1104 ± 47 mg m$^{-2}$ (11.0%). Comparing them with the greatest incidences found on 29 March at the southern stations (ARN and GRA), $\overline{M_{DD}}$ was almost 2 times lower than those found for ARN (in comparison with GRA station, no significant differences were observed). Finally, the $\overline{M_{DD}}$ (daily ftr_M$_{DD}$) at BCN (**Fig. 9a, Table S2**) shows its greatest incidence on 1 April, with values of 478 ± 350

mg m$^{-2}$ (8.7%). Taking as reference the greatest incidence found at the southern stations (on 29 March at ARN and GRA), the $\overline{M_{DD}}$ was almost 4 and 2 times lower than those values observed at ARN and GRA, respectively. No significant differences were found for the daily ftr_M$_{DD}$ between the stations. In overall, for the whole dust event period, the $M_{DD}$ (ftr_M$_{DD}$) values were, on average, 586 ± 602 mg m$^{-2}$ (9.9%) at ARN, 483 ± 385 mg m$^{-2}$ (9.1%) at GRA, 332 ± 185 mg m$^{-2}$ (10.5%) at EVO, 464 ± 365 mg m$^{-2}$ (11.3%) at TRJ and 248 ± 143 mg m$^{-2}$ (9.4%) at BCN (see **Table S2**).

In general, the daily $M_{DD}$ decreased along the dust event as latitude increased, indicating the decay of the dust incidence by overcrossing the dust outbreak through the IP from SW to NE. However, the ftr_M$_{DD}$ did not show significant differences, showing a uniformly well-mixed state of the Dc and Df particles along their pathway over the IP. Therefore, the daily ftr_M$_{DD}$, as well as the ftr_DOD, did not show significant differences among stations (see **Sect. 4.2**). Moreover, the relation between $DOD^{532}$ and $M_{DD}$ are consistent with other studies (e.g. Córdoba-Jabonero et al., 2019, 2021a). Particularly, for low-to-medium

$DOD^{532}$ (<0.30) the $M_{DD}$ values ranged from 82 to 478 mg m$^{-2}$, which are close to those found in the literature (i.e. from 40 to 640 mg m$^{-2}$). Regarding higher $DOD^{532}$ values (>0.30), $M_{DD}$ values were within 542-1809 mg m$^{-2}$, that are in the same range (570 to 2340 mg m$^{-2}$) as in other studies (e. g. Córdoba-Jabonero et al., 2019; 2021a).

The daily-averaged dust CoM height values for the fine and coarse components ($Z_{CoM}^{Df}$ and $Z_{CoM}^{Dc}$) as obtained along the dust intrusion over the five Iberian lidar stations are introduced in **Table 5,** as a measure of the vertical incidence of each dust

component. It is worth highlighting that both $Z_{CoM}^{Df}$ and $Z_{CoM}^{Dc}$ followed the same pattern, confirming the previously assumed well-mixed state between both the Dc and Df particles, independently on the station, with a relative difference between $Z_{CoM}^{Df}$



and $Z_{CoM}^{Dc}$ of around 5% (**Figs. S1 and S2**). Thus, for the sake of simplicity, both the daily-averaged $Z_{CoM}^{Df}$ and $Z_{CoM}^{Dc}$ will be referred to indistinctly as CoM height henceforward in the text.

At the beginning of the dust outbreak, the CoM height was found at low altitudes over each station (~ 2.0 km height). On 27 March at the southern stations, the CoM height ascended (~ 3.0-4.0 km), remaining at that altitude until the end of the maximum dust incidence (29-31 March). From 1 April on, the CoM height gradually lowered until the end of the dust outbreak (~ 1.5-2.5 km, on 5-7 April). At the TRJ station, due to the meteorological conditions, no retrieval was possible on 27 March. However, the CoM height ascended on 28 March from 2.5 km up since the beginning of the episode, reaching ~5 km height, and progressively descended down to ~ 2 km on 5 April. The CoM height at BCN ascended gradually, until reaching 5 km on 31 March (on the day before the greatest incidence). Then, it progressively descended down to ~ 2.5 km height at the end of the dust episode.

Overall, the daily-averaged CoM height increased with latitude from 2.3 km height at ARN to 3.4 km at BCN. For the overall dusty period, a high CoM height standard deviation was found in all stations, being higher than 0.5 km at the southern sites (ARN, GRA, EVO) and ~ 1 km at TRJ and BCN. For comparison with previous studies, the CoM height over central Europe during dust events is observed to vary between 2.3 to 6.6 km (Mona et al., 2006) and from 0.85 to 8 km over south and south-east Europe (Papayannis et al., 2008). For instance, CoM height values of 1-6 and 2-3 km were found at BCN and GRA, respectively (Papayannis et al., 2008; Cazorla et al., 2017).

## 5. Discussion

Since 25 March a low-pressure system (L1) centred over the Canary Islands originated a S/SE CWT, favouring the advection of air masses from Northern Africa to the IP. Days later, another low-pressure system (L2), accompanied by a closed upper-level low-pressure system isolated from the general atmospheric circulation, moved toward the west IP. This configuration, combined with the presence of a diffluent flow at 250 hPa, induced vertical movements that injected dust into the upper troposphere as well as favoured the formation of clouds that prevented some lidar retrievals. On 5 April, L2 had dissipated and northerly flows were verified over the northeast of the IP at higher altitude levels. Thus, this meteorological configuration favoured the injection of a high amount of dust, which still remained suspended over the IP until the end of the dust event.

Due to the meteorological situation each day, dust particles were detected below 3 km a.s.l. at the beginning of the event at the southern stations. Later on, the dust plume ascended up to 8.0 km height due to the atmospheric instability. The greatest dust incidence was observed on 29-31 March, by observing an evident dust layer with a predominance of Dc particles that extended from the surface up to 8.0 km height. In particular, the pronounced dust layer was observed ranging at 2.0-5.0, 3.0-5.0 and 2.0-3.0 km height over ARN, GRA and EVO, with maximum $\beta_p$ peaks of 3.0-8.0, 2.0-6.0 and 2.0-3.0 Mm$^{-1}$ sr$^{-1}$, corresponding to maximal $\beta_{Dc}$ values of 2.0-6.0, 1.0-4.0 and 1.0-3.0 Mm$^{-1}$ sr$^{-1}$, and $\beta_{Df}$ values of 0.5-3.0, 1.0-1.5 and 0.5-1.0 Mm$^{-1}$ sr$^{-1}$, respectively. From 1 April on, the dust plume progressively descended, staying below 3 km height until the end of the episode. Moreover, the dust incidence was weaker with respect to the first part of the dust event, with $\beta_p$ values around 2.5, 1.0 and 2.5 Mm$^{-1}$ sr$^{-1}$ for ARN, GRA and EVO, respectively. Overall, $\delta_p$ ~ 0.30 were found at ARN, being almost constant during the first part of the dust outbreak, decaying until 0.15-0.20 on the final day of the event. Lower $\delta_p$ values around 0.25-0.30 and 0.20-0.30, respectively, were derived at GRA and EVO with respect to those observed at ARN at the beginning of the dust outbreak (around 0.30 on 29-31 March) and decreased down to 0.20 at the end of the episode.

The dust situation at the centre of the IP (TRJ observations) presented the dust plume firstly detected below 4.0 km height, but it immediately ascended up to 10.0 km. The highest incidence was found on 29-31 March, with $\delta_p$ values of 0.25, on average, and maximal $\beta_p$, $\beta_{Dc}$ and $\beta_{Df}$ values ranging 2-18, 1-9, 0.5-3.8 Mm$^{-1}$ sr$^{-1}$, respectively. The dust plume decayed from 8.0 to 4.0 km height from 1 April until the end of the dust episode, and both $\beta_p$ and $\delta_p$ decreased down to 1-3 Mm$^{-1}$ sr$^{-1}$ and 0.15-0.25, respectively.





On the other hand, the dust layer over BCN (northeastern IP) was found between 2.0-3.0 km height on 28 March, when the dust intrusion arrived at the station, showing $\beta_p$ and $\delta_p$ values of 1.0-1.5 Mm$^{-1}$ sr$^{-1}$ and 0.30-0.35, respectively. One day later,

the dust plume showed a particular vertical layered structure. At the beginning of the day, two layers were found at altitudes of 2.0-3.0 and 9.0-10.0 km, with $\beta_p$ peaking around 0.3 and 0.2 Mm$^{-1}$ sr$^{-1}$, respectively, and $\delta_p$ ranging from 0.31 to 0.35 for the first dust layer and ~ 0.25 for the second one. At the end of the day, three layers were observed at altitudes of 1.0-2.0, 4.0-7.0 and 8.0-9.0 km height, with $\beta_p$ peak values of around 0.9, 0.5, and 0.1 Mm$^{-1}$ sr$^{-1}$, respectively, and with $\delta_p$ of 0.33, 0.28 and 0.25, on average. For the successive days, the dust plume followed the same behaviour, reaching its maximum incidence on 1

April, and the intensity was decaying until the end of the episode, with $\beta_p$ and $\delta_p$ values ranging between 0.5-2.0 Mm$^{-1}$ sr$^{-1}$ and 0.20-0.30, respectively.

Dust mass concentrations reached maximal values between 500 and 750 μg m$^{-3}$ at the southern stations (ARN, GRA, EVO) on 29-31 March (the highest dust incidence), being relatively higher (1000 μg m$^{-3}$) at the centre of the IP (TRJ observations) on the same days, and rather lower (around 200 μg m$^{-3}$) at the north-east BCN station on 1 April. It should be noted that, in

general, ftr_$M_{DD}$ was almost constant in altitude with values around 10% at all stations, indicating a well-mixed state between Dc and Df particles. Moreover, it was nearly the same at all the stations, and together with the fact of the reported reduction of the dust incidence with latitude along the overall dusty period, that could confirm that a uniform gravitational settling of both the Dc and Df particles could be produced during their transport across the IP.

By examining the columnar-integrated dust properties, the daily-averaged $DOD^{532}$ (daily-averaged ftr_DOD) showed values

of 1.02 ± 0.26 (25.5%), 0.60 ± 0.04 (28.3%) and 0.36 ± 0.11 (30.6%) at ARN, GRA and EVO stations, respectively, on the days of maximal incidence (29-31 March), meanwhile on the centre of the IP (TRJ observations) maximal values of 0.66 ± 0.03 (28.5%) were observed on 29 March, and reaching $DOD^{532}$ of 0.27 ± 0.20 (25.9%) on 1 April over the North-East of the IP (BCN station). It should be noted that on 1-2 April, the weather conditions prevented lidar observations of the dust intrusion in most stations except at BCN, hence it could be assumed that the dust incidence could have been higher on those days. Later

on (2-7 April), the dust incidence was significantly weaker, being 7, 10 and 3 times lower than the highest dust incidence for the first part of the dust episode (25-31 March) at the southern ARN, GRA and EVO stations, respectively. In general, the mean daily $DOD^{532}$ for the whole dust outbreak decreased by increasing latitude, with a relative difference of 60% from the southern station (ARN) to the north-eastern station (BCN). Regarding the daily fine-to-total $DOD^{532}$ ratio (daily ftr_DOD), similar nearly constant values (~ 25%) were found on the days of maximal dust incidence at each station. The mean daily

ftr_DOD for the overall dust episode did not show significant differences between stations, with values ranging 28-30%, revealing a well-mixed state of the Dc and Df particles along their pathway over the IP.

Moreover, daily $M_{DD}$ ($\overline{M_{DD}}$) (daily ftr_$M_{DD}$) followed the same pattern as the daily $DOD^{532}$, as expected. It progressively increased until reaching values of 1809 ± 396 mg m$^{-2}$ (8.5%), 1018 ± 96 mg m$^{-2}$ (9.8%) and 605 ± 183 mg m$^{-2}$ (10.6%) at the southern Iberian stations (ARN, GRA and EVO), respectively, under the highest dust incidence conditions (29-31 March). The

strongest mass loading is found at ARN, being 1.8 and 3.0 times higher with respect to that found at GRA and EVO, respectively. At the central and NE IP stations (TRJ and BCN), the dust event reached its maximum intensity on 29 March and 1 April, respectively, with $\overline{M_{DD}}$ (daily ftr_$M_{DD}$) values of 1104 ± 47 mg m$^{-2}$ (11.0%) at TRJ and 478 ± 350 mg m$^{-2}$ (8.7%) at BCN, being 1.6 and 3.8 times lower, respectively, than those values observed at ARN. $\overline{M_{DD}}$ (ftr_$M_{DD}$) values on average for the overall dusty period were 586 ± 602 mg m$^{-2}$ (9.9%) at ARN, 483 ± 385 mg m$^{-2}$ (9.1%) at GRA, 332 ± 185 mg m$^{-2}$ (10.5%)

at EVO, 464 ± 365 mg m$^{-2}$ (11.3%) at TRJ and 248 ± 143 mg m$^{-2}$ (9.4%) at BCN. These results show that, on average, the mean $\overline{M_{DD}}$ decreased by latitude increased along the dust intrusion was crossing the IP, as expected. Moreover, almost no differences were observed in the ftr_$M_{DD}$ with latitude increase for the overall dust event, indicating a well-mixed state among Dc and Df particles. Besides, it kept constant (~10%) in time along the dust pathway over the IP, which could confirm that a uniform gravitational settling of both the coarse and fine dust particles could be produced during their transport across the IP.



Due to the particular synoptic situation, the mean CoM height increased with latitude along the overall dust episode, from 2.3 km height at ARN to 3.4 km at BCN, since the dust signature was found at higher altitudes as latitude increased. Maximum daily CoM heights of 3.3, 4.5, 3.2, 4.8 and 5.3 km were found at ARN, GRA, EVO, TRJ and BCN, respectively. In all stations a high CoM height variability is found, being higher than 0.5 km at the southern sites (ARN, GRA, EVO) and around 1 km at TRJ and BCN.

In general, it should be noted that, despite the relatively high intensity of this dust event, the expected ageing of the dust particles (changes in the relative contribution and properties of the coarse and fine dust particles) was barely observed, unlike that was reported in long-range dust events as those reaching central Europe (e.g., Córdoba-Jabonero et al., 2021a).

## 6 Conclusions

An intense dust outbreak arrived at the Iberian Peninsula (IP) in spring 2021 (25 March - 7 April). It was monitored and
analysed by five Iberian lidar stations, strategically positioned to cover mostly the main extension of the Iberian Peninsula (ARN, GRA, EVO, TRJ and BCN). The meteorological conditions determined the particular aerosol scenario. The synoptic situation and back-trajectory analysis revealed that this dust intrusion, which originated in the Saharan region, crossed the Iberian Peninsula from South-West to North-East. Strong dusty conditions were observed from 25 March to 31 March 2021 at the southern ARN, GRA and EVO stations, and from 26 March to 5 April 2021 at the central TRJ and north-eastern BCN
stations. In addition, under the particular synoptic conditions over the Iberian Peninsula, wet deposition occurred at southern and central stations on 1-3 April. From 1 to 7 April dust particles were still found at the southern stations, but in a lesser amount. These particles may have come from the recirculation of air masses moving from NE to SW since the north-east Iberian Peninsula was still under dusty conditions, and also from direct transport from the Sahara (approximately 25-35% of back-trajectories in southern Spain still had their origin in the Sahara region).

The dust vertical distribution was rather homogeneous over the southern (ARN, GRA and EVO) and central (TRJ) stations, whereas the dust plume was substantially more stratified at the northeastern one (BCN). It must be highlighted that the top of the dust plume reached high altitudes over the central and North-East stations (~ 10 km a.s.l.), well above the expected mean height range for mineral dust over Europe. In addition, the mean daily centre-of-mass height for the whole dust episode increased with latitude, showing more variability at TRJ and BCN (standard deviation ~ 1 km overall) than at the southern
stations (standard deviations of 0.5-0.8 km for the whole dust episode). This behaviour was associated with the particular meteorological conditions prevailing at each station during the event, favouring the uplift of dust to the upper troposphere, i.e., the upper troposphere dust belt.

The strongest incidence of the dust outbreak over the Iberian Peninsula was observed on 29-31 March at the southern and central stations (ARN, GRA, EVO, TRJ) and on 1 April at BCN. Dust coarse particles dominated the dusty event, with the
daily fine-to-total mass concentration ratio less than 10 % for the whole event. The mean daily total dust optical depth at 532 nm and total dust mass loading values for the overall dust outbreak decreased as latitude increases. Moreover, only slight differences between the stations were observed in both the fine-to-total dust optical depth at 532 nm and mass loading ratios for the overall dust episode, indicating a well-mixed state among dust coarse and fine particles, which kept nearly constant along the dust pathway over the Iberian Peninsula. A uniform and similar gravitational settling of both coarse and fine particles
may have occurred during their transport across the Iberian Peninsula. Indeed, the expected ageing of the dust particles was barely observed. The Iberian Peninsula is relatively close to the Saharan dust sources and, under certain dust transport conditions, the observations carried out over the Iberian Peninsula do not allow to appreciate any potential ageing processes in the dust particles, unlike that can be observed in long-range dust transport.

This study uses the state-of-the-art two-step POLIPHON algorithm in combination with polarized elastic lidar observations,
in order to separately retrieve aerosol properties for the dust fine and coarse particles. Indeed, their properties can be relatively



well retrieved from lidar systems with polarization capabilities as long as both particle fine and coarse modes have clearly different depolarization ratios.

This work can be very valuable in order to assess the particular radiative (short- and long-wave) effect of both coarse and fine dust particles under intense dusty conditions. This is particularly relevant taking into account the different relative
contributions, on average, of the fine dust particles to the total dust found for their optical properties (30%) with respect to the mass features (10%) along the overall dust event by crossing the IP. Thus, a radiative study of this Saharan dust outbreak will be addressed in the future.

## Data availability

Part of the data used in this publication was obtained as part of the EARLINET network and is publicly available. For additional
lidar data or information, please contact the corresponding author.

## Author contributions

Conceptualization, M-ÁL-C, CC-J, and J-LG-R; methodology, M-ÁL-C, DB-P, MS and F-TC; software, M-ÁL-C and CVC-P; formal analysis, M-ÁL-C, CC-J and J-LG-R; investigation, M-ÁL-C and all authors; resources, CC-J, LA-A, AC, MJC and BA; data curation, M-ÁL-C, CC-J, M-JG-M, AR-G and DB; writing—original draft preparation, M-ÁL-C, CC-J and J-LG-R;
writing—review and editing, M-ÁL-C and all authors; supervision, CC-J and J-LG-R; funding acquisition, CC-J and M-PZ. All authors have read and agreed to the published version of the manuscript.

## Competing interests

The authors declare that they have no conflict of interest.

## Acknowledgements

This research was funded by the Spanish MICINN (PID2020-117825GB-C21, PID2020-120015RB-I00, PID2019-104205GB-C21, PID2019-103886RB-I00, EQC2018-004686-P, CGL2017-90884-REDT, CGL2017-85344-R), the Madrid Regional Government (Y2018/EMT-5177), the Andalusian Regional Government (P20_00136, P18-RT-3820), the University of Granada (A-RNM-430-UGR20) and the EU H2020 programme (GA no. 654109, 778349, 871115 and 101008004). EVO team is co-funded by Portuguese funds through FCT in the framework of ICT (UIDB/04683/2020, UIDP/04683/2020) and
TOMAQAPA (PTDC/CTAMET/29678/2017) projects. ARN team thanks to the technical staff of El Arenosillo site for maintenance and operation support. BCN team particularly thanks J. E. Welton for providing the MPL unit in place at the Barcelona site. The authors thankfully acknowledge the FEDER program for the instrumentation used in this work, which is part of the Laboratorio Singular de Tecnologías Avanzadas Observatorio Global de la Atmósfera (ÁGORA), supported by the University of Granada. J. E. Welton and S. Stewart are warmly acknowledged for their continuous help in keeping the MPL
systems up to date. MALC and CVCP are supported by the INTA predoctoral contract programme. Authors thank NOAA/ESRL Physical Science Laboratory for providing synoptic maps.

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



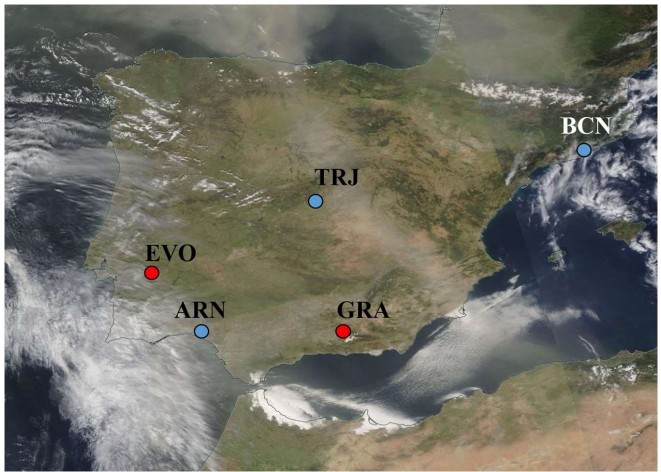

**Figure 1: MODIS image of the corrected reflectance over the Iberian Peninsula on 31 March 2021. The five Iberian lidar stations are marked with a red dot (Raman Lidar, RL) and blue dot (Polarized Micro-Pulse Lidar, P-MPL) (from NE to SW in the Iberian Peninsula): Barcelona (BCN), Torrejón/Madrid (TRJ), Évora (EVO), Granada (GRA), and El Arenosillo/Huelva (ARN) sites.**





820

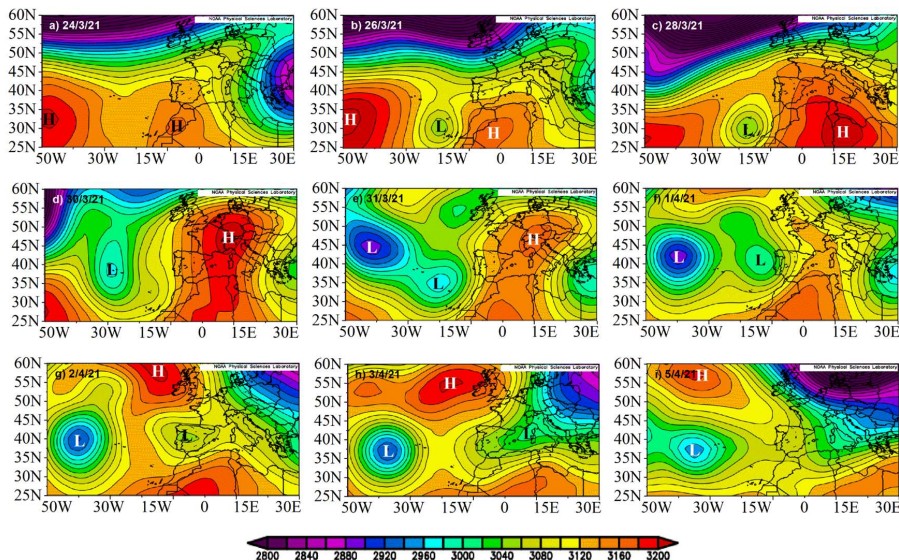

700 hPa Geopotential Height (m) Composite Mean

NCEP/NCAR Reanalysis

**Figure 2: 700 hPa Geopotential Height (in m) Composite Mean for several significant days of the dust outbreak (from left-up to right-down panels): 24, 26, 28, 30 and 31 March 2021 and 1, 2, 3 and 5 April 2021. High- and Low-pressure systems are indicated by H and L, respectively. Images provided by the NOAA/ESRL Physical Sciences Laboratory, Boulder Colorado;**
**http://psl.noaa.gov/; last access: 11 April 2022).**





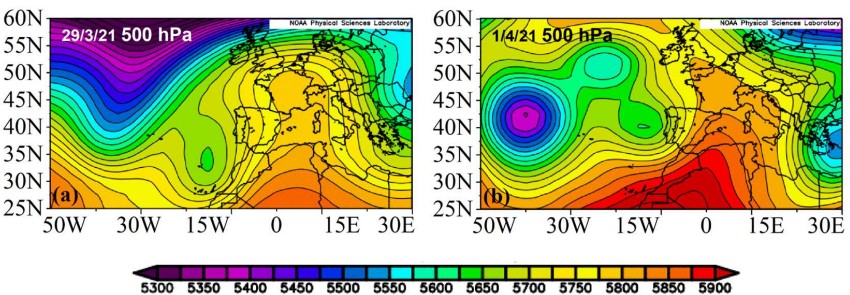

Geopotential Height (m) Composite Mean
NCEP/NCAR Reanalysis

**Figure 3: Geopotential Height (in m) Composite Mean for 29 March 2021 and 1 April 2021 at 500 hPa. Images provided by the NOAA/ESRL Physical Sciences Laboratory, Boulder Colorado (http://psl.noaa.gov/; last access: 11 April 2022).**





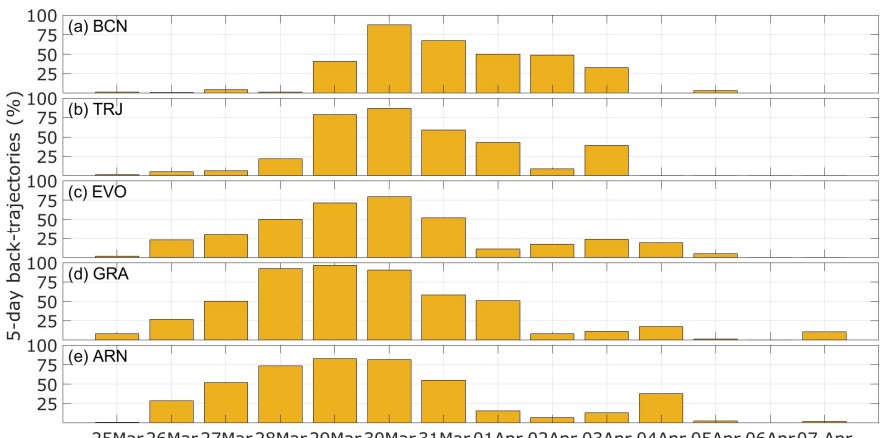


**Figure 4: Percent of HYSPLIT 5-day back-trajectories originated in the Sahara desert area and arrived at each station (from NE to SW, by decreasing latitude): (a) Barcelona (BCN), (b) Torrejón/Madrid (TRJ), (c) Évora (EVO), (d) Granada (GRA) and (e) El Arenosillo/Huelva (ARN).**




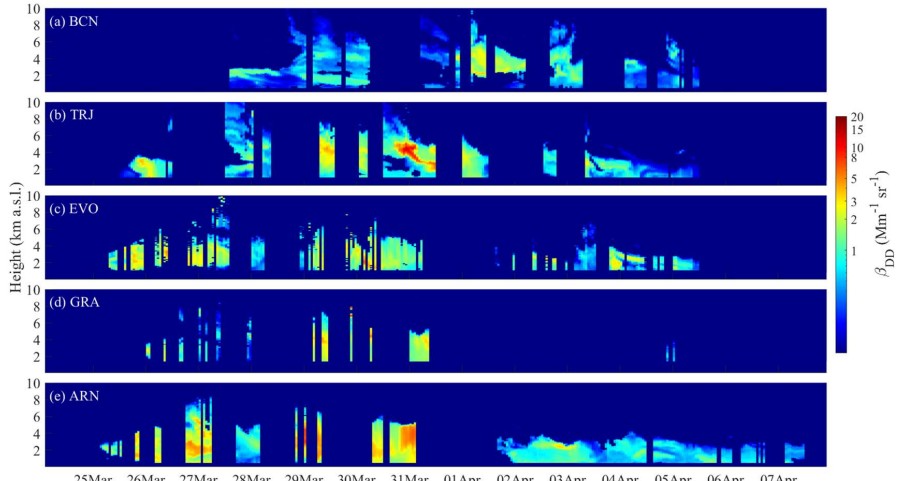

**Figure 5: Temporal evolution of the total dust backscatter coefficient (β$_{DD}$, Mm$^{-1}$ sr$^{-1}$) at the five Iberian lidar stations (from NE to SW, by decreasing latitude): (a) BCN, (b) TRJ, (c) EVO, (d) GRA and (e) ARN. Profile gaps are due to either no possible inversion or no lidar measurements.**


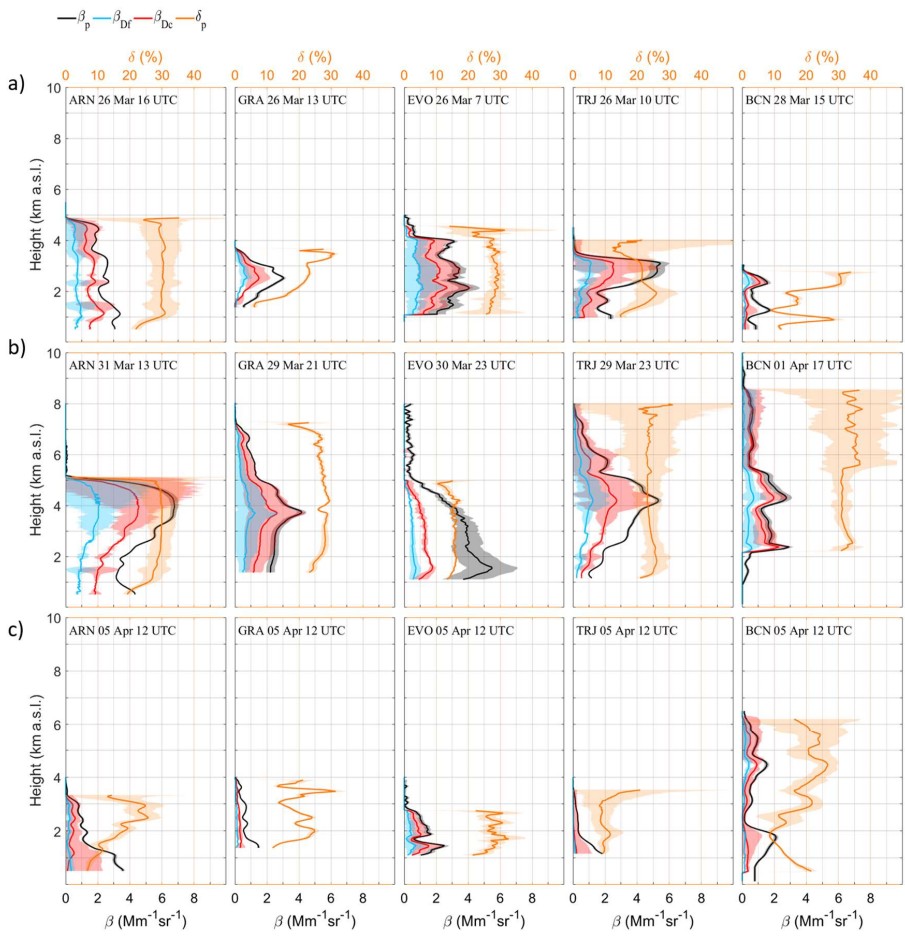

**Figure 6: Optical properties profiles for representative cases (date and time are shown in each panel): (a) beginning of the dust event, (b) maximum incidence and (c) end of the event, observed at (from left to right) ARN, GRA, EVO, TRJ and BCN: the total βp (black) and those separated into dusty components (βDf, blue and βDc, red), together with δp (orange). Errors are shown by shaded bands.**





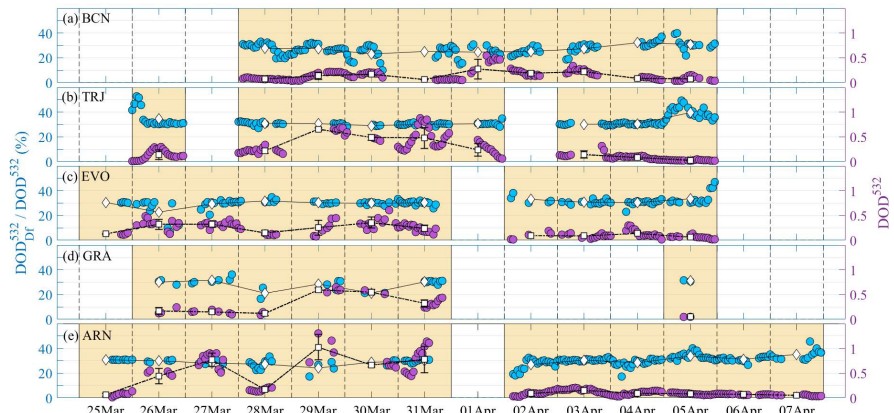

**Figure 7: Temporal evolution of the total dust (DD) optical depth at 532 nm (DOD$^{532}$) (right axis, in purple), and its fine to total ratio (DOD$^{532}_{Df}$/ DOD$^{532}$, %) (left axis, in blue), over the five lidar stations as latitude decreases (from up to down panels): (a) BCN, (b) TRJ, (c) EVO, (d) GRA and (e) ARN. Circles indicate hourly averages. Diamonds and squares indicate daily averages for DOD$^{532}_{Df}$ / DOD$^{532}$ and DOD$^{532}$, respectively.**

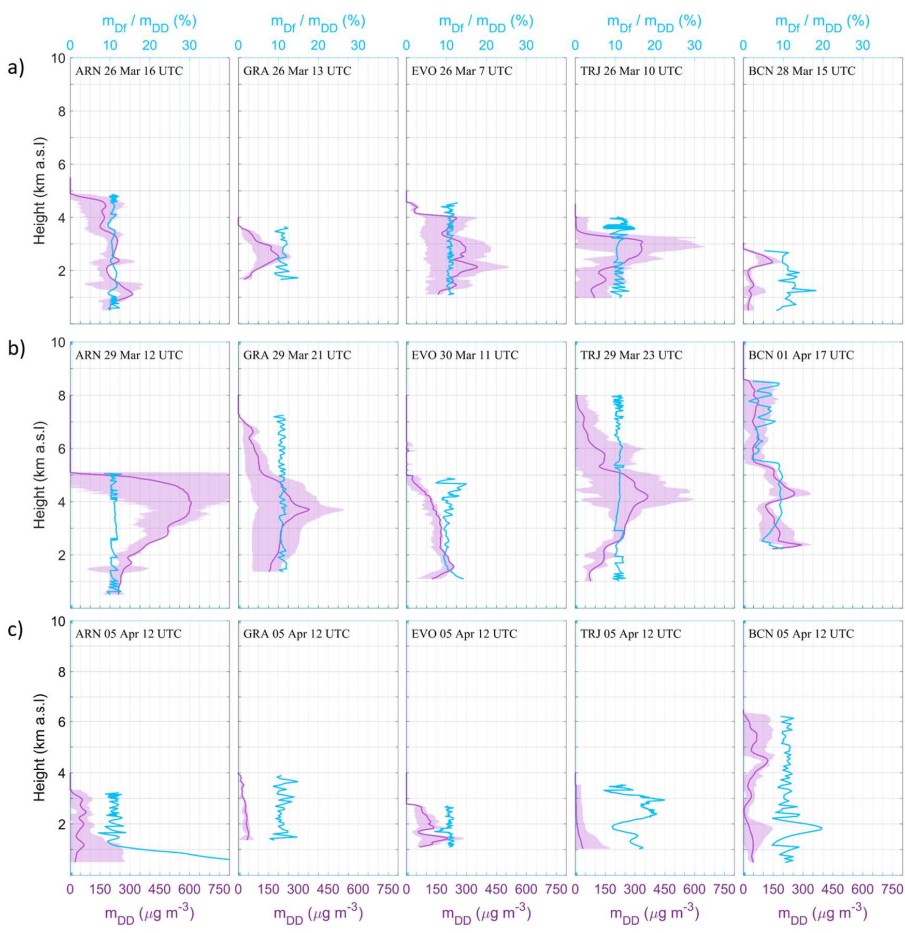

**Figure 8: Dust mass features profiles for representative cases (date and time are shown in each panel): (a) beginning of the dusty event; (b) maximum incidence and (c) end of the event, observed at ARN, GRA, EVO, TRJ and BCN: the total dust mass concentration, $m_{DD}$ (purple), together with its Df/DD ratio, $m_{Df}/m_{DD}$ (%) (blue). Errors are shown by shaded bands.**






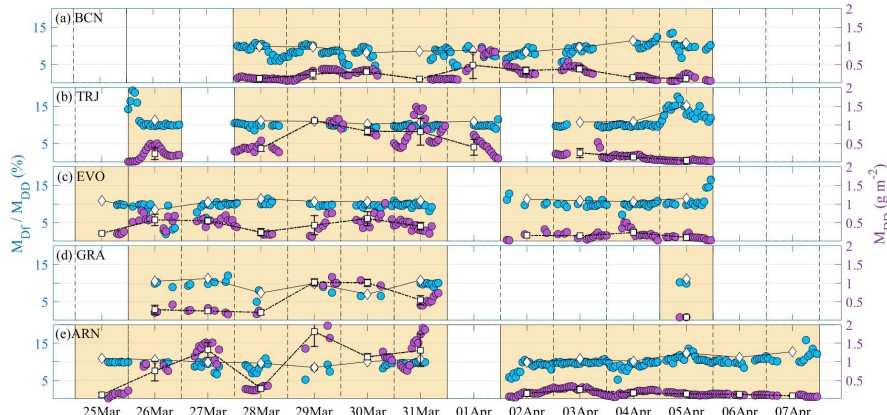

**Figure 9: The same as Fig. 7, but for the relative mass loading of the Df particles ($M_{Df}$, in %) (left axis, in blue) and the total dust mass loading ($M_{DD}$ in g m$^{-2}$) (right axis, in purple).**




**Table 1. Period of the dust outbreak as observed in each station (ordered by increasing latitude).**

| Station | Period |
|---------|--------|
| ARN | 25 March – 7 April |
| GRA | 26 March – 5 April |
| EVO | 25 March – 5 April |
| TRJ | 26 March – 5 April |
| BCN | 28 March – 5 April |




**Table 2. Details of the five Iberian lidar stations used in this work. MPL and RL denote Micro-Pulse Lidar and Raman Lidar, respectively.**

| | | El Arenosillo/ Huelva, Spain (ARN) | Granada, Spain (GRA) | Évora, Portugal (EVO) | Torrejón/ Madrid, Spain (TRJ) | Barcelona, Spain (BCN) |
|---|---|---|---|---|---|---|
| Institution | | Spanish Institute for Aerospace Technology (INTA) | Andalusian Institute for Earth System Research (IISTA-CEAMA). University of Granada | Institute of Earth Sciences (ICT) University of Évora | Spanish Institute for Aerospace Technology (INTA) | Polytechnic University of Catalonia (UPC) |
| Location | | 37.11°N, 6.73°W 40 m a.s.l. | 37.16°N, 3.61°W 680 m a.s.l. | 38.57°N, 7.91°W 293 m a.s.l. | 40.49°N, 3.46°W 568 m a.s.l. | 41.39°N, 2.11°E 125 m a.s.l. |
| Lidar type | | P-MPL | RL | RL | P-MPL | P-MPL |
| Wavelength (nm) | Elastic | 532 | 355, 532, 1064 | 355, 532, 1064 | 532 | 532 |
| | Depolarization | 532 | 532 | 532 | 532 | 532 |
| | Raman | - | 354 and 530 (N2), 407(H20) | 387 and 607 (N2) | - | - |
| Laser source | | Nd:YVO4 | Nd:YAG | Nd:YAG | Nd:YVO4 | Nd:YVO4 |
| Vertical Resolution (m) | | 15 | 7.5 | 30 | 15 | 75 |
| Temporal Resolution (s) | | 60 | 60 | 30 | 60 | 60 |




**Table 3. Assumed values of particle linear depolarization ratio ($\delta_p$), lidar ratio (sr), volume conversion factor (cv in $10^{-12}$ Mm) and particle density ($d_p$ in g cm$^{-3}$) for dust particles.**

| Parameter | Dc | Df | References |
|:---:|:---:|:---:|:---:|
| $\delta_p$ | 0.39 | 0.16 | Mamouri and Ansmann (2017) |
| Lidar Ratio (sr) | 55 | 55 | Ansmann et al. (2019) |
| $c_v$ | 0.83 | 0.23 | Ansmann et al. (2019); Mamouri and Ansmann (2017) |
| $d_p$ (g cm$^{-3}$) | 2.6 | 2.6 | Mamouri and Ansmann (2017) |






**Table 4. Relative uncertainties (%) associated with the lidar-derived particle optical properties and mass features. The sub-indexes refer to p (particle), DD (total dust), Df (fine dust component) and Dc (coarse dust component).**

| Dust Parameter | Symbol | Uncertainty |
|---|---|---|
| Backscatter coefficient ($Mm^{-1}sr^{-1}$) | $\beta_p(z)$ | 10%-15% |
| | $\beta_{DD}(z), \beta_{Df}(z), \beta_{Dc}(z)$ | 10%-30% |
| Particle depolarization ratio | $\delta_p(z)$ | 15%-25% |
| Extinction coefficient ($Mm^{-1}$) | $\sigma_{DD}(z)$ | 20%-30% |
| | $\sigma_{Df}(z)$ | 40%-60% |
| | $\sigma_{Dc}(z)$ | 25%-35% |
| Mass concentration ($g\ m^{-3}$) | $M_{DD}(z)$ | 20%-30% |
| | $M_{Df}(z)$ | 40%-60% |
| | $M_{Dc}(z)$ | 25%-35% |






**Table 5. Daily centre-of-mass height ($Z_{CoM}$ in $km$ a.s.l.) for dust fine (Df) and coarse (Dc) along the particular dust periods for the five lidar stations (ARN, GRA, EVO, TRJ and BCN). The standard deviation values are in brackets.**

| $Z_{COM}$ | | 25 Mar | 26 Mar | 27 Mar | 28 Mar | 29 Mar | 30 Mar | 31 Mar | 1 Apr | 2 Apr | 3 Apr | 4 Apr | 5 Apr | 6 Apr | 7 Apr | 25 Mar − 7 Apr |
|---|---|---|---|---|---|---|---|---|---|---|---|---|---|---|---|---|
| ARN | Dc | 2.2 (0.2) | 2.4 (0.1) | 3.1 (0.3) | 2.9 (0.1) | 3.3 (0.3) | 2.6 (0.1) | 2.9 (0.2) | - | 1.9 (0.5) | 1.8 (0.3) | 1.8 (0.3) | 1.8 (0.2) | 1.3 (0.1) | 1.6 (0.2) | 2.3 (0.7) |
| | Df | 2.2 (0.2) | 2.3 (0.1) | 3.0 (0.5) | 2.5 (0.2) | 3.0 (0.4) | 2.6 (0.1) | 2.9 (0.2) | - | 1.8 (0.4) | 1.8 (0.2) | 1.7 (0.3) | 1.6 (0.1) | 1.3 (0.1) | 1.5 (0.1) | 2.2 (0.6) |
| GRA | Dc | - | 2.6 (0.1) | 4.0 (0.8) | 4.1 (0.6) | 3.7 (0.1) | 4.5 (1.0) | 3.1 (0.1) | - | - | - | - | 2.3 (0.1) | - | - | 3.4 (0.8) |
| | Df | - | 2.6 (0.1) | 3.9 (0.7) | 3.8 (0.6) | 3.7 (0.2) | 4.1 (0.6) | 3.1 (0.1) | - | - | - | - | 2.3 (0.1) | - | - | 3.4 (0.7) |
| EVO | Dc | 2.2 (0.1) | 2.9 (0.4) | 3.5 (0.8) | 3.2 (0.2) | 3.0 (0.3) | 3.0 (0.4) | 2.8 (0.2) | - | 2.4 (0.3) | 2.5 (0.7) | 2.3 (0.3) | 1.8 (0.1) | - | - | 2.7 (0.5) |
| | Df | 2.2 (0.1) | 2.8 (0.3) | 3.4 (0.8) | 3.1 (0.2) | 3.0 (0.3) | 2.9 (0.4) | 2.8 (0.2) | - | 2.3 (0.4) | 2.6 (0.6) | 2.3 (0.3) | 1.8 (0.1) | - | - | 2.7 (0.5) |
| TRJ | Dc | - | 2.2 (0.6) | - | 4.8 (0.8) | 4.0 (0.1) | 3.7 (0.1) | 4.0 (0.8) | 2.7 (0.3) | - | 2.8 (0.3) | 2.2 (0.2) | 1.9 (0.2) | - | - | 3.1 (1.0) |
| | Df | - | 2.2 (0.6) | - | 4.7 (0.7) | 4.0 (0.1) | 3.7 (0.2) | 4.0 (0.8) | 2.7 (0.3) | - | 2.8 (0.3) | 2.1 (0.2) | 1.9 (0.4) | - | - | 3.1 (1.0) |
| BCN | Dc | - | - | - | 2.3 (0.3) | 3.8 (0.6) | 3.2 (0.3) | 5.3 (0.8) | 4.1 (0.5) | 3.3 (0.3) | 3.3 (0.8) | 2.5 (0.2) | 2.6 (0.9) | - | - | 3.4 (0.9) |
| | Df | - | - | - | 2.1 (0.3) | 4.1 (0.7) | 3.2 (0.4) | 5.6 (0.9) | 4.3 (0.4) | 3.5 (0.3) | 3.2 (0.7) | 2.4 (0.2) | 2.4 (0.8) | - | - | 3.4 (1.1) |
