# Peer review of "Vertical characterization of the dust fine and coarse particles during an intense Saharan dust outbreak over the Iberian Peninsula in springtime 2021"

_Atmospheric Chemistry and Physics, 2022_

## Author Response (AR1)

The authors would like to thank the reviewers for their comments that definitively have improved the article. For a better clarity, the authors' response can be found next, marked with different colours: blue for reviewer #1, and purple for reviewer #2.

**Response of authors to the reviewer's comments #1**

The paper has high technical quality and meets the magazine's requirements. The methodology is well described and the results presented satisfy the proposed objectives. I recommend the following technical corrections:

- Figure 2: Please, change the color of the text in the pictures. It is not possible to read them. Done. Figure 2 has been modified. The labels have been improved to a better clarity (line 715).
- Figure 3: Please, use the same pattern of the font size applied in the previous figure. Done. Figure 3 has been modified to follow the same pattern as the previous figure (line 715). Moreover, for the sake of simplicity, Figure 2 and Figure 3 have been merged into Figure 2.
- Figure 5: The color bar sounds strange. Are the numbers in the correct position? The color-map of Figure 5 (now Figure 3) has been modified for better clarity.

**Response of authors to the reviewer's comments #2**

Review of the paper by Lopez-Cayuela et al.

The paper presents in detail a Saharan dust event that was observed over the entire Iberian Peninsula from a number of lidar stations. Emphasis is given in the separation of fine and course dust profiles, applying the well established two-step POLIPHON algorithm. Additionally estimates of the corresponding mass concentration profiles of fine and coarse dust are made using conversion factors from previous studies. The paper is well structured and the results presented, as stated in the conclusions, are important for studying the relatively different radiative effects of fine and coarse dust. The paper could be accepted for publication in ACP but the authors should put more effort in a revised version to highlight the main findings and their relevance. The current version of the paper is too lengthy and detailed concerning the presentation of the measurements and the synoptic situation and thus at the end it is not clear which are the major findings of the paper. Therefore, I would suggest the authors would consider the following general comments before their paper would be accepted.

 The authors should mention early in the manuscript, why this event was interesting. There are already similar studies for the same area. What is the new approach here? What is the new information from these measurements compared to the previous studies? Then the discussion and description should provide the answers to these questions.

Following the reviewer's recommendations, the interest in the work has been highlighted in the introduction (lines 87-95):

"This work aims to study an exceptionally intense and long-lasting Saharan dust event, occurring over the IP in springtime 2021 and monitored by five Iberian lidar stations, which covered mostly the IP from SW to NE. Other authors also studied intense and extreme dust events of the IP in several stations, but just focusing on the optical properties of the event (Cazorla et al., 2017; Salvador et al., 2019). The new approach of this work is not only focused to apply a validated methodology to separate the contribution of both the coarse and fine dust particles, but also to include a study of the evolution of their microphysical properties along their transport from the SW to the NE IP in terms of the estimation of their mass concentrations, among other features. Moreover, thanks to the wide spatial coverage for dust monitoring, it is possible to assess the potential ageing of the dust particles along their transport by crossing the IP, by examining the minor changes found in the contribution and properties of the coarse and fine dust particles."

2. Sections 4.2 and 4.3 are too lengthy and the reader misses this way the important information. The authors want to show the time and latitudinal evolution of the event through the lidar data analysis. At the end it is not very clear what is the answer, i.e is there a significant change with time and latitude?

Authors thank the reviewer to have pointed out this issue. Sections 4.2 and 4.3 has been revised, modified and rewritten for a better reading and understanding, without missing any scientific result and conclusion. Moreover, Section 4.2 has been split into Sections 4.1.2 and 4.2.2, and Section 4.3 has been also split into Sections 4.3.1 and 4.3.2, for clarity.

Additionally, the text of Section 4.1 has also been restructured for clarity.

All those modifications (related to this study of the dust evolution on time and latitude) are explicitly written in the text (see paragraphs below), in addition to Figures 5 and 7 and Tables S1 and S2 of the Supplementary Material. Figures and Tables have been renumbered.

Moreover, in order to present more clearly the results to the reader and improve thus the manuscript, it has been added a table (Table 2 in the new version of manuscript) with the averaged values of the aerosol optical depth at 532nm, mass loading and centre-of-mass height for the whole dust event period.

Lines 323-327: "In summary,  $\overline{DOD^{532}}$  decreased as latitude increased, with mean values ranging from 0.34 ± 0.35 at ARN to 0.14 ± 0.08 at BCN station, i.e. 40% lower in the NE with respect to SW of the IP; however, the ftr\_DOD did not show significant differences (remained around 29%) between SW and NE stations (see **Table 2**). These results indicate the decay of the dust incidence by overcrossing through the IP from SW to NE regions, together with a uniformly well-mixed state of the Dc and Df particles along that pathway."

Lines 385-389: "In summary, for the whole dust event period, the MDD (ftr\_MDD) values were, on average, 586 ± 602 mg m-2 (9.9%) at ARN, 483 ± 385 mg m-2 (9.1%) at GRA, 332 ± 185 mg m-2 (10.5%) at EVO, 464 ± 365 mg m-2 (11.3%) at TRJ and 248 ± 143 mg m-2 (9.4%) at BCN (see **Table 2**). Thus, the daily  $M_{DD}$  decreased along the dust event as latitude increased, being a 75% lower in BCN with respect to ARN station. However, significant differences were unobserved for the ftr\_MDD among stations, likewise found for the ftr\_DOD (see **Sect. 4.2.2**)."

Lines 400-402: "In summary, the mean CoM height increased with latitude along the overall dust episode, from 2.3 km height at ARN to 3.4 km at BCN, since the dust signature was found at higher altitudes as latitude increased (see **Table 2**)."

**3. The discussion section has a lot of repetitions with the previous section.**

The Results and Discussion sections have been merged, and duplicities have been corrected and/or removed just in case. The manuscript has been reduced a 15% (approximately 2000 words).

4. The authors mention ageing of dust. They should mention in the discussion what are the metrics they use for potentially identifying ageing processes? Changes in concentration, depol ratio, lidar ratio etc.?

It has been explicitly written in the text of the new version of the manuscript:

Line 300: "The ftr\_DOD is one of the proxies used in this work to study the ageing of dust." Line 370: "Note that ftr\_MDD is a proxy used in this work to study the ageing of dust."

5. Time-Latitude cross sections of average quantities (e.g. from intensive parameters, fine/coarse dust, ratio, center of mass etc), would help to have an overview the evolution of the event.

In the new version of the Supplementary Material, Tables S1 and S2 correspond to a timelatitude cross-section of the daily Dust Optical Depth at 532 nm (DOD532) and Mass Loading ( $M_L$ ) for fine dust (Df), coarse dust (Dc) and total dust (DD) along the particular dusty periods for the five Iberian lidar stations (ARN, GRA, EVO, TRJ and BCN). Moreover, Figures 5 and Figure 7 show a time-latitude cross of the total dust optical depth at 532 nm and mass loading (and their fine-to-total ratio), respectively, over the five lidar stations.

Regarding the Reviewer's comment, for clarification, the Table 2 is added, showing the mean  $DOD^{532}$  and  $M_L$  for fine dust (Df), coarse dust (Dc), total dust (DD) and Df-to-total dust ratio (Df/DD) for the whole dust episode at the five lidar stations (ARN, GRA, EVO, TRJ and BCN).